# Learning interacting dynamical systems with latent Gaussian process ODEs

**Çağatay Yıldız**[*]
University of Tübingen
cagatay.yildiz@uni-tuebingen.de

**Melih Kandemir**
University of Southern Denmark
kandemir@imada.sdu.dk

**Barbara Rakitsch**
Bosch Center for Artificial Intelligence
barbara.rakitsch@de.bosch.com

## Abstract

We study uncertainty-aware modeling of continuous-time dynamics of interacting objects. We introduce a new model that decomposes independent dynamics of single objects accurately from their interactions. By employing latent Gaussian process ordinary differential equations, our model infers both independent dynamics and their interactions with reliable uncertainty estimates. In our formulation, each object is represented as a graph node and interactions are modeled by accumulating the messages coming from neighboring objects. We show that efficient inference of such a complex network of variables is possible with modern variational sparse Gaussian process inference techniques. We empirically demonstrate that our model improves the reliability of long-term predictions over neural network based alternatives and it successfully handles missing dynamic or static information. Furthermore, we observe that only our model can successfully encapsulate independent dynamics and interaction information in distinct functions and show the benefit from this disentanglement in extrapolation scenarios.

## 1   Introduction

A broad spectrum of dynamical systems consists of multiple interacting objects. Since their interplay is typically *a priori* unknown, learning interaction dynamics of objects from data has become an emerging field in dynamical systems [1, 2, 3]. The ever-growing interest in interaction modeling is due to the diversity of real-world applications such as autonomous driving [4], physical simulators [5], and human-robot interactions [6]. Standard time-series algorithms or deep learning approaches (e.g. recurrent neural networks), that have been designed for single-object systems, do not scale to a large number of interacting objects since they do not exploit the structural information of the data.

In recent years, graph neural networks (GNNs) have emerged as a promising tool for interactive systems, where objects are represented as graph nodes. State-of-the-art methods learn interactions by sending messages between objects in form of multi-layer perceptrons [1] or attention modules [7]. These methods yield highly flexible function approximators that achieve accurate predictions when trained on large-scale datasets. However, their predictions come without calibrated uncertainties, hindering their reliable implementation for uncertainty-aware applications.

In contrast, Gaussian processes (GPs) are well-known for providing calibrated uncertainty estimates. They have been successfully employed on discrete time-series data [8, 9, 10] and, more recently,

---

[*]Work performed during internship at Bosch Center for Artificial Intelligence

36th Conference on Neural Information Processing Systems (NeurIPS 2022).

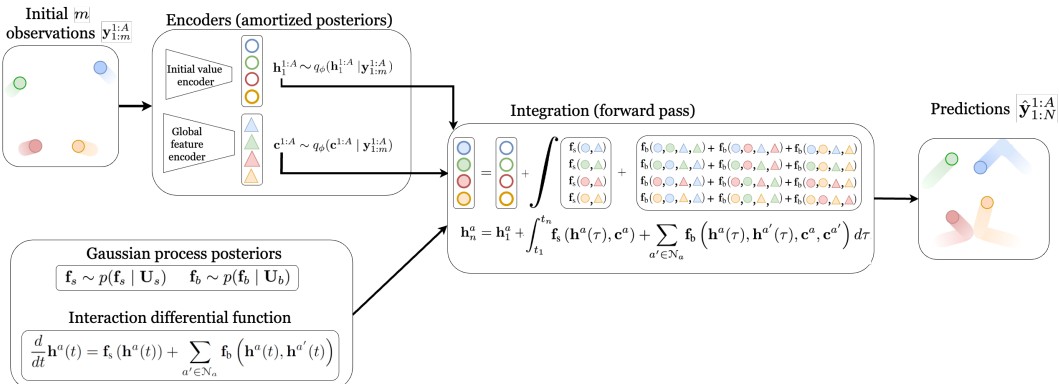

Figure 1: An overview of our predictive model. *(Top-left)* An input bouncing balls sequence with four balls, which move independently other than collision (interaction) times. An encoder is used to extract initial values and global latents. *(Bottom-left)* The differential function is formed by sampling from the GP posteriors on the independent kinematics and interaction functions. *(Middle&right)* Given the samples, predicted trajectories are computed using the forward integration of the differential function.

to continuous-time generalizations of these methods [11, 12, 13]. Importantly, none of the these works adresses dynamical models for interacting systems. While it is possible to study each object in isolation, ignoring the interaction effects might lead to inaccurate predictions.

In this work, we address the shortcomings of both model families by presenting an uncertainty-aware continuous-time dynamical model for interacting objects. Our formulation decomposes the dynamics into independent (autonomous) and interaction dynamics. While the former governs the motion of an object in isolation, the latter describes the effects that result from interactions with neighboring objects. For successful uncertainty characterization, we propose to infer the unknown independent and interaction dynamics by two distinct GPs. We demonstrate that having a function-level GP prior on the individual dynamics components is the key to successfully disentangling these dynamics, which in turn allows for interpretable predictions and leads to improved extrapolation behavior.

We employ latent Gaussian process ordinary differential equations (GP-ODEs) for dynamics learning, allowing to learn complex relationships between interacting objects without the need of having access to fully observed systems. Thanks to recently proposed decoupled sampling scheme [14], the computational complexity of our model scales linearly with the number of time points at which the ODE system is evaluated. As a result, our algorithm scales gracefully to datasets with thousands of sequences. To demonstrate the benefits of our framework, we exhaustively test our method on a wide range of scenarios varying in function complexity, signal-to-noise ratio, and system observability. Our model consistently outperforms non-interacting dynamical systems and alternative function approximators such as deterministic/Bayesian neural networks.

## 2 Background

In this section, we give background on continuous-time systems and Gaussian processes. Both together form the backbone of our uncertainty-aware framework for interactive dynamical systems.

### 2.1 Continuous-time Dynamical Systems

Continuous-time dynamical systems are often expressed using differential functions $\dot{\mathbf{x}}(t) \equiv \frac{d}{dt}\mathbf{x}(t) \equiv \mathbf{f}(\mathbf{x}(t))$, where $\mathbf{x}(t) \in \mathbb{R}^D$ represents the state of an ODE system at time $t$ and and $\mathbf{f} : \mathbb{R}^D \mapsto \mathbb{R}^D$ is the *time differential function* governing the dynamics evolution. The state solution $\mathbf{x}(t_1)$ at an arbitrary time $t_1$ is characterized by the initial value at time point $t_0$ and the differential function:

$$\mathbf{x}(t_1) = \mathbf{x}(t_0) + \int_{t_0}^{t_1} \mathbf{f}(\mathbf{x}(\tau)) \, d\tau.$$

Existing work aims to approximate the unknown differential by Gaussian processes [15, 11] or neural networks [16]. These methods have shown to accurately capture the dynamics and outperform

their discrete-time counterparts in a wide range of applications such as time series forecasting [17], classification [18] or reinforcement learning [19]. Furthermore, ODE models allow to easily inject domain knowledge into the system, enabling interpretable and flexible hybrid models [20, 21, 22].

## 2.2 Gaussian Processes

Gaussian processes (GPs) define priors over functions [23]:

$$f(\mathbf{x}) \sim \mathcal{GP}(\mu(\mathbf{x}), k(\mathbf{x}, \mathbf{x}')),$$

where $f : \mathbb{R}^D \to \mathbb{R}$ maps $D$-dimensional inputs into one-dimensional outputs. GPs are fully specified in terms of their mean and their covariance:

$$\mathbb{E}[f(\mathbf{x})] = \mu(\mathbf{x}), \qquad \operatorname{cov}[f(\mathbf{x}), f(\mathbf{x}')] = k(\mathbf{x}, \mathbf{x}'),$$

where $\mu : \mathbb{R}^D \to \mathbb{R}$ is the mean and $k : \mathbb{R}^D \times \mathbb{R}^D \to \mathbb{R}$ is the kernel function. GPs can be treated as an extension of a multivariate normal distribution to infinitely many dimensions, where any fixed set of inputs $\mathbf{X} \in \mathbb{R}^{N \times D}$ follows the Gaussian distribution

$$p(\mathbf{f}) = \mathcal{N}(\mathbf{f} \mid \boldsymbol{\mu}_{\mathbf{X}}, \mathbf{K}_{\mathbf{XX}}), \tag{1}$$

where the mean function $\boldsymbol{\mu}_{\mathbf{X}}$ is evaluated at inputs $\mathbf{X}$, and $\mathbf{K}_{\mathbf{XX}}$ the kernel function evaluated at all input pairs in $\mathbf{X}$. While GPs provide a natural mechanism to handle uncertainties, their computational complexity grows cubically with the number of inputs. This problem is often tackled by *sparse GPs*, which rely on augmenting the GP with $M$ inducing inputs $\mathbf{Z} = [\mathbf{z}_1^T, \ldots, \mathbf{z}_M^T]^T \in \mathbb{R}^{M \times D}$ and corresponding output variables $\mathbf{u} = [u_1, \ldots, u_M]^T \in \mathbb{R}^{M \times 1}$ with $u_m \equiv f(\mathbf{z}_m)$ [24, 25]. Assuming the commonly used zero-mean prior, the conditional distribution over $f(\mathbf{X})$ follows the GP:

$$p(\mathbf{f} \mid \mathbf{u}) = \mathcal{N}(\mathbf{f} \mid \mathbf{K}_{\mathbf{XZ}} \mathbf{K}_{\mathbf{ZZ}}^{-1} \mathbf{u}, \ \mathbf{K}_{\mathbf{XX}} - \mathbf{K}_{\mathbf{XZ}} \mathbf{K}_{\mathbf{ZZ}}^{-1} \mathbf{K}_{\mathbf{ZX}}), \tag{2}$$

where $\mathbf{K}_{\mathbf{ZZ}}$ is the covariance between all inducing points $\mathbf{Z}$, and $\mathbf{K}_{\mathbf{XZ}}$ between the input points $\mathbf{X}$ and the inducing points $\mathbf{Z}$. The inducing points can thereby be interpreted as a compressed version of the training data in which the number of inducing points $M$ acts as a trade-off parameter between the goodness of the approximation and scalability.

In this work, we employ the squared exponential kernel $k(\mathbf{x}, \mathbf{x}') = \sigma^2 \exp\left(-\frac{1}{2} \sum_{d=1}^D \frac{(x_d - x_d')^2}{\ell_d^2}\right)$, where $x_d$ denotes the $d$-th entry of the input $\mathbf{x}$, $\sigma^2$ is the output variance and $\ell_d$ is the dimension-wise lengthscale parameter.

## 3 Interacting Dynamical Systems with Latent Gaussian Process ODEs

In Sec. 3.1, we describe our continuous-time formulation for systems of interacting objects. It decomposes the dynamics into independent kinematics and an interaction component that takes the interactions to neighboring objects into account. Placing a GP prior over the individual components is essential in order to arrive at *(i)* calibrated uncertainty estimates and *(ii)* disentangled representations as we later on also verify in our experiments. In Sec. 3.2, we embed the GP dynamics into a latent space that can accomodate missing static or dynamic information. Both together allows the application of our continuous-time formulation to a wide range of scenarios and allows for learning interpretable dynamics. We conclude this section by our variational inference framework (Sec 3.3) based on sampling functions from GP posteriors.

### 3.1 Interacting Dynamical Systems

We assume a dataset of $P$ sequences $\mathcal{Y} = \{\mathbb{Y}_1, \ldots, \mathbb{Y}_P\}$, where each sequence $\mathbb{Y} \equiv \mathbf{Y}_{1:N} \equiv \mathbf{y}_{1:N}^{1:A}$ is composed of measurements of $A$ objects at time points $\mathcal{T} = \{t_1, \ldots, t_N\}$. Without loss of generality, we assume that the measurement $\mathbf{y}^a(t_n) \equiv \mathbf{y}_n^a \in \mathbb{R}^O$ is related to the physical properties of object $a$, such as position and velocity, which can routinely be measured by standard sensors. The *dynamic state* of object $a$ at any arbitrary time $t$ is denoted by a latent vector $\mathbf{h}^a(t) \in \mathbb{R}^D$, which does not necessarily live in the same space as the observations. We furthermore assume that each object $a$ is associated with a *global* feature vector $\mathbf{c} \in \mathbb{R}^C$, which corresponds to the static attributes that remain constant over time. Finally, we denote the concatenation of all states by $\mathbf{H}(t) = [\mathbf{h}^1(t), \ldots \mathbf{h}^A(t)] \in \mathbb{R}^{A \times D}$ and all globals by $\mathbf{C} = [\mathbf{c}^1, \ldots \mathbf{c}^A] \in \mathbb{R}^{A \times C}$.

In the following, we propose to disentangle the complex continuous-time dynamics into *independent kinematics* and *interaction* differentials. More concretely, we introduce the following dynamics:

$$\frac{d}{dt}\mathbf{H}(t) = \left[\frac{d}{dt}\mathbf{h}^1(t), \ldots, \frac{d}{dt}\mathbf{h}^A(t)\right], \tag{3}$$

$$\frac{d}{dt}\mathbf{h}^a(t) = \mathbf{f}_\mathrm{s}\left(\mathbf{h}^a(t), \mathbf{c}^a\right) + \sum_{a' \in \mathcal{N}_a} \mathbf{f}_\mathrm{b}\left(\mathbf{h}^a(t), \mathbf{h}^{a'}(t), \mathbf{c}^a, \mathbf{c}^{a'}\right), \tag{4}$$

where $\mathcal{N}_a$ denotes the set of neighbors of object $a$ in a given graph. The first function $\mathbf{f}_\mathrm{s} : \mathbb{R}^{D+C} \mapsto \mathbb{R}^D$ models the independent (autonomous) effects, which specifies how the object would behave without any interactions. The second function $\mathbf{f}_\mathrm{b} : \mathbb{R}^{2D+2C} \mapsto \mathbb{R}^D$ models the *interactions* by accumulating messages coming from all neighboring objects. Since message accumulation is the *de-facto* choice in interaction modeling [2, 26], the additive form of the differential function is a very generic inductive bias.

Our formulation models interactions between pairs of objects explicitly via the differential equation (Eq. (5)). Higher-order interactions are taken into account via the continuous-time formulation that allows information to propagate through the complete graph via local messages over time such that the state of the object $\mathbf{h}_n^a$ can also depend on objects that are not directly connected in the graph. In contrast to discrete formulations, for which the message passing speed is limited by the sampling rate, our continuous-time formulation enjoys instant propagation of information across objects. Finally, please see Section for an investigation of our interaction component under a kernel perspective.

**Remark-1**   In Sec **??** we demonstrate two straightforward extensions of our formulation with non-linear message accumulation, which we empirically show to have no gain over our formulation.

**Remark-2**   Previous GP-based ODE methods [15, 11, 12] assume a black-box approximation to the unknown system $\frac{d}{dt}\mathbf{H}(t) = \mathbf{f}(\mathbf{H}(t))$ whereas our state representation gracefully scales to a varying number of objects and also include global features.

## 3.2   Probabilistic Generative Model

Real-world data of interactive systems necessitates embedding the dynamics into a latent space in order to allow for missing information; the observations may contain only partial estimates of the states and the globals $\mathbf{C}$ might not be observed at all. We account for those circumstances by treating the states $\mathbf{H}$ and the globals $\mathbf{C}$ as latent variables, leading to following generative model (see Figure 1):

$$\mathbf{h}_1^a \sim \mathcal{N}(\mathbf{0}, \mathbf{I}), \quad \mathbf{c}^a \sim \mathcal{N}(\mathbf{0}, \mathbf{I}),$$
$$\mathbf{f}_\mathrm{s}(\cdot) \sim \mathcal{GP}(\mathbf{0}, k_\mathrm{s}(\cdot, \cdot)), \quad \mathbf{f}_\mathrm{b}(\cdot) \sim \mathcal{GP}(\mathbf{0}, k_\mathrm{b}(\cdot, \cdot)),$$
$$\mathbf{h}_n^a = \mathbf{h}_1^a + \int_{t_1}^{t_n} \mathbf{f}_\mathrm{s}\left(\mathbf{h}^a(\tau), \mathbf{c}^a\right) + \sum_{a' \in \mathcal{N}_a} \mathbf{f}_\mathrm{b}\left(\mathbf{h}^a(\tau), \mathbf{h}^{a'}(\tau), \mathbf{c}^a, \mathbf{c}^{a'}\right) d\tau, \tag{5}$$
$$\mathbf{y}_n^a \sim p(\mathbf{y}_n^a | \mathbf{h}_n^a),$$

where we introduced a standard Gaussian prior over the initial latent state, and assumed that the data likelihood decomposes across time and objects. We furthermore model unknown functions $\mathbf{f}_\mathrm{s}$ and $\mathbf{f}_\mathrm{b}$ under independent vector-valued GP priors.

In our experiments, we further set $p(\mathbf{y}_n^a | \mathbf{h}_n^a) = \mathcal{N}(\mathbf{y}_n^a | \mathbf{B}\mathbf{h}_n^a, \mathrm{diag}(\boldsymbol{\sigma}_e^2))$, where $\mathbf{B} \in \mathbb{R}^{O \times D}$ maps from the latent to the observational space and $\boldsymbol{\sigma}_e^2 \in \mathbb{R}_+^O$ is the noise variance. We further fix $\mathbf{B} = [\mathbf{I}, \mathbf{0}]$ where $\mathbf{I} \in \mathbb{R}^{O \times O}, \mathbf{0} \in \mathbb{R}^{O, D-O}$, in order to arrive at an interpretable latent space in which the first dimensions correspond to the observables. This assumption is fairly standard in the GP state-space model literature since more complex emission models can be subsumed in the transition model without reducing the model complexity [27].

Modeling partially observed systems often leads to non-identifiability issues that hamper optimization and ultimately lead to deteriorated generalization performance. One way to counteract this behavior is to  inject prior physical knowledge into the system by decomposing the state space of each object $\mathbf{h}^a(t) \equiv [\mathbf{s}^a(t), \mathbf{v}^a(t)]$ into *position* $\mathbf{s}^a(t)$ and *velocity* $\mathbf{v}^a(t)$ components [28]. Using elementary physics, the differential function has then the form of $\frac{d}{dt}\mathbf{h}^a(t) = \left[\mathbf{v}^a(t), \frac{d}{dt}\mathbf{v}^a(t)\right]$ with $\frac{d}{dt}\mathbf{v}^a(t) = \mathbf{f}_\mathrm{s}\left(\mathbf{h}^a(t), \mathbf{c}^a\right) + \sum_{a' \in \mathcal{N}_a} \mathbf{f}_\mathrm{b}\left(\mathbf{h}^a(t), \mathbf{h}^{a'}(t), \mathbf{c}^a, \mathbf{c}^{a'}\right).$

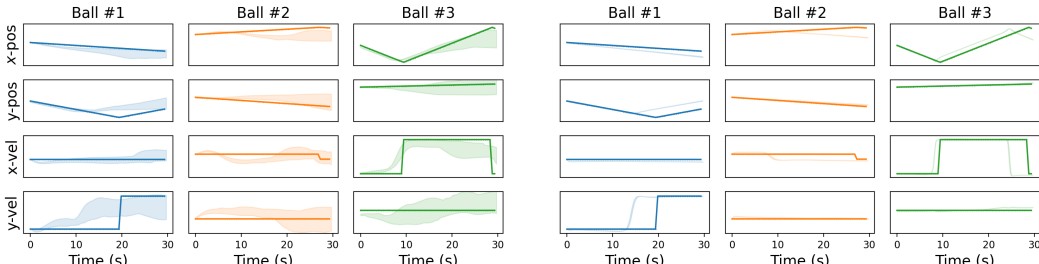

Figure 2: Test predictions for I-GPODE (left) and I-NODE (right) on bouncing balls dataset. The solid curves are the groundtruth trajectories and the shaded regions denote the predicted $95\%$ confidence intervals. I-GPODE (ours) yields better calibrated long-term predictions than I-NODE. Additional results can be found in Figure **??**.

**Remark** Unlike previous work, our formulation incorporates global features $\mathbf{c}$ that modulate the dynamics. In many applications such as control engineering and reinforcement learning, the dynamics are modulated by external control signals [29] which can also be incorporated into our framework.

### 3.3 Variational Inference

Next, we derive an efficient approximate inference scheme that provides a high level of accuracy and model flexibility. In the above formulation, the model unknowns are the initial values $\mathbf{H}_1 \equiv \mathbf{h}_1^{1:A}$, global variables $\mathbf{C} = \mathbf{c}^{1:A}$ and the differentials $\mathbf{f}_s$ and $\mathbf{f}_b$. Since the exact posterior $p(\mathbf{f}_s, \mathbf{f}_b, \mathbf{H}_1 \, \mathbf{C} | \mathcal{Y})$ in non-linear ODE models is intractable, we opt for stochastic variational inference [30]. We first describe the form of the approximate posterior and then discuss how to optimize its parameters.

**Variational family** Similarly to previous work [16], we resort to amortized inference for initial values and global variables, $\mathbf{H}_1 \sim q_\phi(\mathbf{H}_1 | \mathbf{Y}_{1:N})$ and , $\mathbf{C} \sim q_\rho(\mathbf{C} | \mathbf{Y}_{1:N})$, where $\phi$ is the parameters of a neural network encoder that outputs a Gaussian distribution with diagonal covariance. For the unknown time differentials $\mathbf{f} = \{\mathbf{f}_s, \mathbf{f}_b\}$, we follow the standard sparse GP approximation in which each output dimension $d \in [1, D]$ has its own independent set of inducing values $\mathbf{U}_{s,d}, \mathbf{U}_{b,d} \in \mathbb{R}^D$ and kernel output variances $\sigma_{s,d}^2, \sigma_{b,d}^2 \in \mathbb{R}_+$. Inducing locations $\{\mathbf{Z}_s, \mathbf{Z}_d\}$ and lengthscales $\{\boldsymbol{\ell}_s, \boldsymbol{\ell}_b\}$ are shared across output dimensions. Denoting the collection of inducing points by $\mathbf{U} = \{\mathbf{U}_s, \mathbf{U}_b\}$, we arrive at the mean-field variational posterior:

$$q(\mathbf{U}) = \prod_{d=1}^{D} \mathcal{N}(\mathbf{U}_{s,d} \mid \mathbf{m}_{s,d}, \mathbf{S}_{s,d}) \mathcal{N}(\mathbf{U}_{b,d} \mid \mathbf{m}_{b,d}, \mathbf{S}_{b,d}),$$

where the means $\{\mathbf{m}_{s,d}, \mathbf{m}_{b,d}\}_{d=1}^{D}$ and the covariances $\{\mathbf{S}_{s,d}, \mathbf{S}_{b,d}\}_{d=1}^{D}$ are free variational parameters. Putting everything together, our variational approximation becomes as follows [31]:

$$q(\mathbf{H}_1, \mathbf{C}, \mathbf{f}, \mathbf{U}) \equiv q(\mathbf{H}_1) q(\mathbf{C}) p(\mathbf{f}_s | \mathbf{U}_s) p(\mathbf{f}_b | \mathbf{U}_b) q(\mathbf{U}),$$

where $p(\mathbf{f}_s | \mathbf{U}_s)$ and $p(\mathbf{f}_b | \mathbf{U}_b)$ follow Eq. (2). Our variational family makes two assumptions that are fairly standard in the (deep) GP literature (e.g. [32]): *(i)* we apply the same independence assumptions in the approximate posterior as in the prior resulting in a mean-field solution, and *(ii)* we assume that the inducing outputs $\mathbf{U}$ capture the sufficient statistics of the training data allowing the use of the prior $p(\mathbf{f} \mid \mathbf{U})$ in the approximate posterior.

**Variational bound** We then seek to optimize the parameters of the approximate posterior $q$ by maximizing a lower bound to the evidence [30]:

$$\log p(\mathcal{Y}) \geq \int q(\mathbf{H}_1, \mathbf{C}, \mathbf{f}, \mathbf{U}) \log \frac{p(\mathcal{Y}, \mathbf{H}_1, \mathbf{C}, \mathbf{f}, \mathbf{U})}{q(\mathbf{H}_1, \mathbf{C}, \mathbf{f}, \mathbf{U})} d\mathbf{H}_1 d\mathbf{C} d\mathbf{f} d\mathbf{U}.$$

In the following, we detail its computation for a single data instance $\mathbf{Y}_{1:N}$, omitting its generalization to multiple sequences for the sake of better readability,

$$\log p(\mathbf{Y}_{1:N}) \geq \mathbb{E}_q \left[\log p(\mathbf{Y}_{1:N} | \mathbf{H}_1, \mathbf{C}, \mathbf{U})\right] - \text{KL}[q(\mathbf{H}_1) || p(\mathbf{H}_1)] - \text{KL}[q(\mathbf{C}) || p(\mathbf{C})] - \text{KL}[q(\mathbf{U}) || p(\mathbf{U})], \tag{6}$$

where KL denotes the Kullback–Leibler divergence.

**Likelihood computation via decoupled sampling from GP posteriors**  Computing the conditional log-likelihood $\log p(\mathbf{Y}_{1:N}|\mathbf{H}_1, \mathbf{C}, \mathbf{f}, \mathbf{U})$ entails a forward pass in time (Eq. (5)) which can be done with any standard ODE solver. The difficulty lies in marginalizing over the approximate posterior of the initial latent states $q(\mathbf{H}_1)$, global variables $q(\mathbf{C})$, and the GP functions $q(\mathbf{f}, \mathbf{U})$. Each marginalization step alone is already analytically intractable, let alone their combination. We therefore opt for Monte Carlo integration which gives us an unbiased estimate of the expected log-likelihood. We start by drawing $L$ samples from the approximate posteriors

$$\mathbf{H}_1^{(l)} \sim q_\phi(\mathbf{H}_1|\mathbf{Y}_{1:N}), \quad \mathbf{C}^{(l)} \sim q_\rho(\mathbf{C}|\mathbf{Y}_{1:N}), \quad \mathbf{U}^{(l)} \sim q(\mathbf{U}), \quad \mathbf{f}^{(l)}(\cdot) \sim p(\mathbf{f}|\mathbf{U}), \qquad (7)$$

where $l$ denotes the sample index and $\mathbf{f}^{(l)}(\cdot)$ is a function drawn from the sparse GP posterior. Sampling from the GP posterior naively scales cubically with the number of data points. Moreover, since we do not know a-priori on which points the ODE solver evaluates the function, we would have to sequentially draw points from the posterior. While this can still be done cubically in time by performing low-rank updates, it often leads to numerical instabilities for small step sizes. To overcome this challenge, we resort to the decoupled sampling scheme proposed in [14], where we first draw the prior samples from a set of random Fourier features and then update them using Matheron's rule to obtain posterior samples. After having sampled the quadruple via Eq. (7), we can compute the trajectory $\mathbf{H}_{1:N}^{(l)}$ deterministically by forward integrating Eq. (5). Monte Carlo estimate of the log-likelihood becomes

$$\mathbb{E}_q\left[\log p(\mathbf{Y}_{1:N}|\mathbf{H}_1, \mathbf{C}, \mathbf{f}, \mathbf{U})\right] \approx \frac{1}{L} \sum_{l,n,a} \log p(\mathbf{y}_n^a|\mathbf{h}_n^{a^{(l)}}),$$

where the log-likelihood term decomposes between objects and between time points, enabling doubly stochastic variational inference [33].

**KL regularizer**  The prior distributions over the inducing variables follow the Gaussian process $p(\mathbf{U}) = p(\mathbf{U}_s)p(\mathbf{U}_b)$ with $p(\mathbf{U}_s)$ and $p(\mathbf{U}_b)$ following the equivalent of Eq. (1). Since we furthermore assumed a standard Gaussian prior $p(\mathbf{H}_1) = \mathcal{N}(\mathbf{0}, \mathbf{I})$ and $p(\mathbf{C}1) = \mathcal{N}(\mathbf{0}, \mathbf{I})$ with suitable dimensions for the initial values, all KL terms can be computed in closed form.

**Computational complexity**  Evaluating the differential function costs $O(M^2(A + I))$ for a graph with $A$ objects, $I$ interactions and $M$ inducing points. The term $O(M^2 A)$ stems from the independent kinematics, the term $O(M^2 I)$ from the interaction effects. As a consequence, forward integration using RK4 solver takes $O(TM^2(A + I))$ time, where $T$ is the number of time points.

### 3.4 Comparison to Standard GPODEs

Our approach enhances the capabilities of the GPODE model family in the following three aspects:

**Explicit modeling of interactions**  Standard GPODEs model interactions by allowing the time differential to take the whole state vector $\mathbf{H}(t)$ as input and to learn one independent GP for each latent dimension. This shared latent space assumption entails three major drawbacks: *(i)* obligation to fix the object count as a model hyperparameter, *(ii)* dependency of the learned model on a predefined ordering of the objects in the scene, *(iii)* inevitable growth of the latent dimensionality proportional to the object count. The latter sets a severe bottleneck especially for GP modeling as the performance of many kernel functions in widespread use are highly sensitive to input dimensionality. (For example, on the bouncing ball dataset with $N_a$ balls and $D$ latent states per object, GPODE needs to learn a latent function with $N_a D$-dimensional inputs and outputs.) In contrast, I-GPODE needs only to learn two functions, the independent kinematics $\mathbf{f}_s$ and the interaction function $\mathbf{f}_b$, whose input sizes scale independently of $N_a$. Our Table 2 indicates that learning interaction dynamics without the strong inductive bias of our model is difficult and the GPODE model chooses to stay at the prior instead leading to deteriorated MSEs and ELLs.

**Disentangled representation**  We infer object-specific latent variables that modulate the dynamics, which allows our model to disentangle the dynamics from static object properties (e.g. charge information). The interpretability and likely physical correspondence of the disentangled factors have the potential to facilitate the use of our approach in transfer learning and explainable AI applications.

**Inference of latent state dynamics**  We perform the learning in a latent space, where the initial value of a trajectory is given by an encoder, leading to the following two advantages: First, the

Table 1: A taxonomy of the state-of-the-art ODE systems with NN/GP dynamics, with/without interactions, and with/without latent dynamics. These methods also form the baselines we compare with [neural ODE (NODE), Bayesian NODE (BNODE), latent dynamics (L), interacting (I)].

| Function uncertainty | Interacting | Latents | Reference idea | Abbreviation |
|:---:|:---:|:---:|---|---|
| ✗ | ✗ | ✗ | Neural ODE [16] | NODE |
| ✗ | ✗ | ✓ | Latent NODE [17] | NODE-L |
| ✗ | ✓ | ✗ | GDE [34] | I-NODE |
| ✗ | ✓ | ✓ | LG-ODE [35] | I-NODE-L |
| ✓ | ✗ | ✗ | Bayesian ODE [11] | GPODE |
| ✓ | ✗ | ✓ | GP-SDE [13] | GPODE-L |
| ✓ | ✗ | ✓ | ODE$^2$VAE [28] | BNODE-L |
| ✓ | ✓ | ✗ | Our work | I-GPODE |
| ✓ | ✓ | ✓ | Our work | I-GPODE-L |

dynamical system and the data points may live in different spaces, which facilitates learning from high-dimensional sequences. Second, Bayesian modeling of state dynamics on a latent space enables reliable quantification and principled treatment of sources of uncertainty, such as imprecision of modeling assumptions, approximation error, and measurement noise.

## 4 Related Work

**GPs for ODEs**  GPs for modeling ODE systems have been studied in a number of publications (e.g. [11, 12, 13, 15, 36]). With the notable exception of [13], they only consider systems in which the dynamics are defined in the data space. The work that is closest to ours from a technical perspective is [11, 12] that also employ decoupled sampling in order to compute consistent trajectories during inference. We are not aware that interacting dynamical systems under a GP prior have been studied previously, either in the continuous or in the discrete time setting.

**Neural ODEs for dynamical data**  Since the debut of Neural ODEs (NODEs) [16], much progress has been made on how to model sequential data in the continuous time domain using neural networks. These works [17, 37] assume a black-box approximation to unknown ODE systems $\frac{d}{dt}\mathbf{H}(t) = \mathbf{f}(\mathbf{H}(t))$, where $\mathbf{f}$ is a deterministic neural network, and the latent space is separated from the observational space using a neural network decoder. A subtle difference in our approach is that we decided to use linear mapping instead. However, when the outputs are high-dimensional, e.g. video data, this can be easily changed. Few works have integrated function uncertainty into NODEs by putting a prior over the neural network weights [28, 38]. To the best of our knowledge, none of these works addressed interacting systems. However, we still compare against an interactive adaptation of these methods.

**Modeling interacting dynamics**  Interacting dynamical systems have first been considered for discrete time-step models and for the deterministic setting [1, 26, 39]. Many of these discrete formulations can be transferred to the continuous case as shown in [3, 34, 40]. This also holds true for our approach, for which the discretized version of the dynamics (Eq. (3), (4)) can be easily cast into one of the existing frameworks (e.g [39]). These works have also been extended to the probabilistic context using a variational auto-encoder [2, 35]. The hidden variables are thereby used to either encode the initial latent states or static information. In contrast to our work, none of these approaches allow for function uncertainty in the dynamics. Finally, [41] proposes a symbolic physics framework, differing from our method in its search-based fixed grammar describing the dynamics.

Finally, we provide a summary of related techniques and derived comparison partners for our experiments in Table 1.

## 5 Experiments

We compare our approach against state-of-the-art methods in a large number of scenarios that differ in function complexity, signal-to-noise ratio, and observability of the system. The empirical findings

suggest that our model leads to improved reliability of long-term predictions while being able to successfully encapsulate autonomous dynamics from interactions effects. In all our experiments, we use the RK4 ODE solver and ACA library [42] for integration and gradient computation. Due to the space limit, we refer to the Supplementary Material for more detailed information about the experimental setup and comparison methods (see also Table 1 for an overview). Our PyTorch [43] implementation can be found in https://github.com/boschresearch/iGPODE (GNU AGPL v.3.0 license).

## 5.1 Experiment Details

**Datasets**   We illustrate the performance of our model on two benchmark datasets: *bouncing balls* [44] and *charges* [2]. These datasets involve freely moving $N_a = 3$ balls that collide with each other and $N_a = 5$ particles that carry randomly assigned positive or negative charges leading to attraction or repulsion, respectively. All simulations are performed in frictionless, square boxes in $2D$. We generate 100 bouncing balls training sequences with different levels of Gaussian perturbations to simulate measurement noise (see Supplementary Section **??** for details). Since the charges dataset requires inferring the charge information, we use 10k train sequences without observation noise as in [2] and similarly use 500 training sequences when velocity information is missing.

**Partial observability**   Prior works typically evaluate their methods on datasets with position and velocity observations [1, 2]. However, having access to the full state of an object is for many real-world problems unrealistic, e.g. many tracking devices can measure positions, but cannot measure velocity or acceleration. To test the model performance in such scenarios, we consider an additional bouncing balls dataset in which the velocities are not observed. On the charges dataset, we assume that position and velocity are observed, but treat the charge information as missing.

**Reported metrics**   We quantify the model fit by computing the expected log-likelihood (ELL) of the data points under the predictive distribution. Further, we report the mean squared error (MSE) between ground truth and predictions over all predicted time steps and over all objects (see Supplementary Section **??** for the exact definitions). Each experiment is repeated five times and we report the mean and standard deviation of both metrics on test sequences.

## 5.2 Empirical Findings

Due to our latent variable construction, ODE state dimensionality $D$ could be arbitrary even though the observations are four-dimensional. To choose an appropriate state dimensionality, we study if the model performance can be improved by augmenting the state space with auxiliary dimensions ($D > 4$). Table **??** shows that increasing the model flexibility beyond need leads to overfitting as we observe lower training but significantly higher test error. Consequently, if not stated otherwise, we use a four-dimensional latent space for each object that corresponds to position and velocity in $x$ and $y$ coordinates and observations consist of their noisy versions. Next, we discuss the main findings.

### 5.2.1 Interacting dynamics are superior over standard formulation

We consider three *bouncing ball* datasets with varying noise levels to reflect different levels of problem difficulties. To demonstrate the merits of our decomposed formulation in Eq. (4), we compare it against a standard, GP-based non-interacting dynamical model (GPODE) in which the time differential takes the whole state vector $\mathbf{H}(t)$ as input [11]. As shown in Table 2, our interaction model consistently outperforms its standard counterpart irrespective of the noise level and function approximator. We also note that the results are consistent when we replace the GP approximation with deterministic and Bayesian neural networks (NNs), which indicates the robustness of the inductive bias. (see Table **??**).

Table 2: Comparison of standard and interacting GP-ODE systems on bouncing ball datasets.

| NOISE LEVEL | MODEL | MSE ↓ | ELL ↑ |
|---|---|---|---|
| NO NOISE | I-GPODE | **17.3 ± 1.2** | **−25.9 ± 3.6** |
| | GPODE | 23.9 ± 0.9 | −597.6 ± 28.5 |
| LOW NOISE | I-GPODE | **17.8 ± 0.9** | **−26.5 ± 4.5** |
| | GPODE | 23.6 ± 0.7 | −580.8 ± 19.8 |
| HIGH NOISE | I-GPODE | **18.4 ± 0.9** | **−29.4 ± 1.1** |
| | GPODE | 24.5 ± 1.0 | −569.3 ± 27.5 |

### 5.2.2 GP approximation yields more calibrated uncertainties

Next, we compare our GP interaction model (I-GPODE) against neural network variants with/without uncertainty (I-NODE, I-BNODE), obtained by replacing GPs with NNs and Bayesian NNs. The findings in Table 3 strongly indicate that I-GPODE learns better calibrated uncertainties, measured by ELL, than I-BNODE and I-NODE (see Figure 2 and **??** for visual illustrations). In terms of MSE, we observe that I-NODE tends to attain the smallest values. However, increasing noise levels deteriorates the performance of I-NODE and eventually the results become comparable to I-GPODE. The results stay consistent when we repeat the experiment with two balls (see Table **??**).

Table 3: Comparison of our GP approximation against alternative function approximators.

| NOISE LEVEL | MODEL | MSE ↓ | ELL ↑ |
|---|---|---|---|
| NO NOISE | I-GPODE | $17.3 \pm 1.2$ | $\mathbf{-25.9 \pm 3.6}$ |
| | I-NODE | $\mathbf{8.5 \pm 0.9}$ | $-61.9 \pm 8.6$ |
| | I-BNODE | $20.4 \pm 0.6$ | $-37.8 \pm 4.2$ |
| LOW NOISE | I-GPODE | $17.8 \pm 0.9$ | $\mathbf{-26.5 \pm 4.5}$ |
| | I-NODE | $\mathbf{13.3 \pm 0.9}$ | $-96.2 \pm 12.5$ |
| | I-BNODE | $21.0 \pm 0.8$ | $-36.4 \pm 2.8$ |
| HIGH NOISE | I-GPODE | $18.4 \pm 0.9$ | $\mathbf{-29.4 \pm 1.1}$ |
| | I-NODE | $\mathbf{15.6 \pm 1.5}$ | $-128.9 \pm 23.2$ |
| | I-BNODE | $20.8 \pm 0.7$ | $-47.6 \pm 10.5$ |

### 5.2.3 GP approximation enables disentangled function learning

In the following, we study whether the estimated functions can disentangle independent kinematics from interaction effects. We train I-GPODE, I-NODE and I-BNODE on a dataset with three balls and evaluate the trained independent dynamics function $\mathbf{f}_s$ on a test dataset with one ball (since the test dataset incorporates a single object, the dynamics do not involve the interaction function $\mathbf{f}_b$). Three test sequences as well as the independent dynamics function predictions are illustrated in Figure 3 (see Table **??**

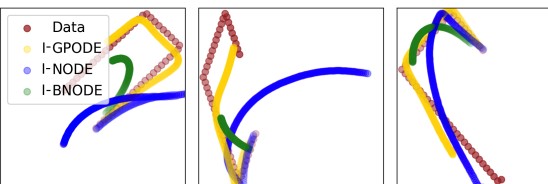

Figure 3: Disentanglement between independent kinematics and interactions. Our model variants are trained on a bouncing ball dataset with three balls and tested with a single ball using the independent dynamics function $\mathbf{f}_s$ only. We show the mean predictions with circles darkened as time lapses.

for a quantitative comparison and Figure **??** for additional illustrations). We observe that I-NODE predictions tend to deviate from the test trajectory more quickly compared to I-GPODE predictions. We conjecture that this behaviour is because neural networks are overflexible and thus the learned functions may not necessarily decompose independent kinematics from interaction effects whereas the function-level regularization of I-GPODE helps with disentanglement.

### 5.2.4 Structured dynamics improve latent dynamics learning

In our last bouncing ball experiment, we move to a setting in which the velocities are no longer observed. First, we keep the velocities as latent states and contrast two variants of our model, i.e. with structured latent space (I-GPODE-L-S) and with unstructured latent space (I-GPODE-L). As shown in Table 4, injecting strong prior knowledge helps in this challenging setting in which the first order model clearly fails (see also Table **??** for training results). Finally, we compare I-GPODE-L-S with I-GPODE, which drops the velocity component from the latent space (hence learning the dynamics

Table 4: Comparison of I-GPODE variants on a bouncing ball dataset without velocity observations. The suffices "-L" and "-S" stand for latent variable model and structured state space.

| NOISE | MODEL | MSE ↓ | ELL ↑ |
|---|---|---|---|
| NO NOISE | I-GPODE | $24.1 \pm 1.2$ | $-300 \pm 23$ |
| | I-GPODE-L | $23.3 \pm 0.8$ | $-176 \pm 301$ |
| | I-GPODE-L-S | $\mathbf{18.2 \pm 1.0}$ | $\mathbf{-33 \pm 2}$ |
| LOW NOISE | I-GPODE | $23.3 \pm 0.8$ | $-239 \pm 11$ |
| | I-GPODE-L | $47.0 \pm 47.4$ | $-761 \pm 922$ |
| | I-GPODE-L-S | $\mathbf{20.5 \pm 0.9}$ | $\mathbf{-76 \pm 22}$ |
| HIGH NOISE | I-GPODE | $23.8 \pm 0.7$ | $-254 \pm 16$ |
| | I-GPODE-L | $119.3 \pm 94.3$ | $-920 \pm 982$ |
| | I-GPODE-L-S | $\mathbf{21.8 \pm 0.7}$ | $\mathbf{-115 \pm 19}$ |

in the data space). As demonstrated in Table 4, I-GPODE is clearly outperformed by I-GPODE-L-S. It can thus be suggested that our latent variable construction is necessary in presence of missing states.

### 5.2.5 Global latent variables boost performance

In the final part of the experiments, we consider a more challenging dataset of charged particles. Since the dynamics are modulated by unknown charges, we turn to our global latent variable formulation. In other words, our learning task becomes simultaneously inferring the functions $\mathbf{f}_s$ and $\mathbf{f}_b$, the initial latent states $\mathbf{H}_1$, as well as a latent variable $\mathbf{c}^a \in \mathbb{R}$ associated with each observed trajectory $\mathbf{y}_{1:N}^a$. To form the upper and lower performance bounds, we include two baselines in which the charges are either observed or completely dropped from the model description. The results are shown in Table 5. We notice that the structured

Table 5: Comparison of interacting GPODE and NODE systems on the charges dataset (no noise).

|  | MODEL | MSE $\downarrow$ | ELL $\uparrow$ |
|---|---|---|---|
| WITHOUT GLOBAL LATENTS | I-GPODE | $19.2 \pm 0.9$ | $-171 \pm 7$ |
|  | I-NODE | $14.1 \pm 0.8$ | $-460 \pm 36$ |
| WITH GLOBAL LATENTS | I-GPODE | $15.4 \pm 2.3$ | $-172 \pm 88$ |
|  | I-NODE | $9.9 \pm 0.6$ | $-282 \pm 14$ |
|  | I-GPODE-S | $12.9 \pm 2.1$ | $\mathbf{-177 \pm 60}$ |
|  | I-NODE-S | $\mathbf{9.7 \pm 0.2}$ | $-282 \pm 8$ |
| GLOBALS OBSERVED | I-GPODE | $10.7 \pm 1.1$ | $-97 \pm 9$ |
|  | I-NODE | $7.5 \pm 1.2$ | $-148 \pm 27$ |

state space formulation boosts the performance of I-GPODE. However, the effect is less pronounced compared to the previous setting in which dynamic information is missing. See Table **??** for more results with different global variable encoders.

## 6 Discussion

We have presented the first uncertainty-aware model for continuous-time interacting dynamical systems. By embedding the dynamics into a latent space, our formulation yields a flexible model family that can be applied to a variety of different scenarios. In our experiments, we found that our approach leads to improved disentanglement and superior calibration of long-term predictions.

Exploring useful applications of our disentangled representation is also an interesting direction for future research. Accurate identification of independent kinematics and interaction effects could enable useful downstream functionalities. For instance, one can perform algorithmic recourse [45] by counterfactual interventions at the object or interaction level.

**Modeling limitations** The capacity of our GP is limited by the choice of the kernel, e.g., the RBF kernel assumes that the dynamics are stationary. While our model formulation can be combined with arbitrary kernel functions and it is possible to increase the kernel expressiveness, e.g by building composite kernels or coming up with hand-crafted features, these approaches are often time-consuming and lead to highly parameterized kernels that are difficult to learn.

**Approximation errors** Our posterior inference scheme is inaccurate due to our variational framework, approximation errors that accumulate in time during future prediction, and numerical errors caused by numerical integration of ideally continuous dynamics and its solution.

**Broader impact** In our work, we propose a methodological contribution which is blind to specific data distributions. Its potential and unforeseeable side-effects in fairness-sensitive or safety-critical applications need to be investigated in a dedicated study.

## Acknowledgements

The Bosch group is carbon neutral. Administration, manufacturing and research actvities do not longer leave a carbon footprint. This also includes GPU clusters on which the experiments have been performed. Cagatay Yildiz is funded by the Deutsche Forschungsgemeinschaft (DFG, German Research Foundation) under Germany's Excellence Strategy – EXC-Number 2064/1 – Project number 390727645. We would like to thank Jakob Lindinger and Michael Herman for discussions and proofreading.

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
