# OpenReview forum: "Learning interacting dynamical systems with latent Gaussian process ODEs"
_NeurIPS.cc/2022/Conference — NeurIPS 2022 Accept_

### Official Review · Reviewer_gdCd · 2022-06-20

**Rating:** 4
**Confidence:** 3
**Soundness:** 3 good
**Presentation:** 3 good
**Contribution:** 1 poor

**Summary:**

The paper takes a Bayesian approach to model dynamical systems with Gaussian processes and combines it with a message passing modeling of the dynamics.

**Questions:**

Can you show that the functions you learn $f_s,f_b$ learn the write functions in the sense that if you disable the interactions then $f_b$ alone would return the correct dynamics?

**Strengths And Weaknesses:**

Strengths:
- The presentation is pretty clear.
- The method makes sense as it combines two well established approaches.

Weaknesses:
- Limited novelty. It is a relatively small modification of GPODE.
- Limited significance. I-NODE achieves better results so main gain is calibration which is a small gain.
- Results are presented in a wasteful, table 2 and 3 should be joint together for better clarity.
- The claim in the first line in the abstract that "for the first time uncertainty-aware modeling of continuous-time dynamics of interacting objects" is false. E.g. Xu et al "A Bayesian-Symbolic Approach to Reasoning and Learning in Intuitive Physics".

Small remark: you have $p(Y_{1:N}|\cdot)$ in eq. 6 which isn't clear

---

> ### Author Response · Authors · 2022-08-02
> **Our response to reviewer comments**
>
>
>
> - **Comment:** *"Limited novelty. It is a relatively small modification of GPODE."*
> -  **Our response:** Please see the "novelties" paragraph in "comments to all reviewers".
>
> $ $
>
>
> - **Comment:** *"Limited significance. I-NODE achieves better results so main gain is calibration which is a small gain."*
> -  **Our response:** Please see the "MSE" paragraph in "comments to all reviewers".
>
> $ $
>
>
> - **Comment:** *"Results are presented in a wasteful, table 2 and 3 should be joint together for better clarity"*
> -  **Our response:** Thanks for the feedback, we were also considering to have a long table instead.
> However, each table conveys a different message and merging them makes it more difficult
> to communicate the individual results. This is why we would kindly prefer to keep them
> separate.
>
> $ $
>
>
> - **Comment:** *"The claim in the first line in the abstract that "for the first time uncertainty-aware modeling of
> continuous-time dynamics of interacting objects" is false. E.g. Xu et al "A Bayesian-Symbolic
> Approach to Reasoning and Learning in Intuitive Physics""*
> -  **Our response:** Thanks for bringing this paper into our awareness, we will add it to the
> paragraph "Modeling interacting dynamics" in our related work section. Although being
> relevant, our approach differs from the paper in certain respects: *(i)* they consider a fixed
> grammar describing the dynamics, *(ii)* they perform a search within the grammar to "learn"
> the dynamics while we propose black-box function approximation to be learned by gradient
> descent, *(iii)* for forward integration, they use a simple Euler approach, which effectively
> yields a discrete-time method, while we utilize the more accurate Runge-Kutta method of
> fourth order.
>
> $ $
>
>
> - **Comment:** *"Small remark: you have $p(Y_{1:N} |\cdot)$ in eq. 6 which isn’t clear"*
> -  **Our response:** We will replace it by $p(Y_{1:N}|  \mathbf{H}_1, \mathbf{C}, \mathbf{f}, \mathbf{U} )$.
>
> $ $
>
>
> - **Question:** *"Can you show that the functions you learn $f_s,f_b$ learn the write functions in the sense that if you disable the interactions then $f_b$ alone would return the correct dynamics?"*
> -  **Our response:** Our main goal in Section 5.2.3 is to show which function approximator (GP, NN, BNN) disentangles individual effects from interactions the best.
>         For this, we trained our system on a dataset with three objects and tested on a bouncing ball system with one object. Due to the lack of interactions in the test setup, ODE solutions are computed solely by the independent kinematics function $\mathbf{f}_s$. Predicted trajectories in Figure 3 justify our GP approximation as it clearly captures the dynamics.

---

> > ### Comment · Reviewer_gdCd · 2022-08-03
> > **NA**
> >
> > - Regarding novelty, I thought the encoder-decoded was part of GPODE so the novelty is higher than I initially thought but not by much. Also, I don't see any ablation study that shows its effect (especially when all the experiments are low dimensional).
> >
> > - Regarding MSE and ELL: Uncertainty is important, but coming at the expense of predictive accuracy is problematic. Also, we can re-calibrate to correct overconfident deterministic predictions (see Levi et al "Evaluating and Calibrating Uncertainty Prediction in Regression Tasks" for regression calibration and issues with Kuleshov et al). It would be more convincing if you show better ELL results after re-calibrating I-NODE.
> >
> > - Regarding the "for the first time..." - I did not say you were similar to this work, but that the claim is wrong and should be removed.

---

> > > ### Author Response · Authors · 2022-08-07
> > > **.**
> > >
> > > - We thank the reviewer for appreciating our latent variable construction that allows us (i) embedding the dynamics in latent space and (ii) inferring object-specific latents that modulate the dynamics.
> > > We kindly note that we perform ablation studies for both scenarios in our paper:
> > > (i) In Section 5.2.4, we compare the performance of different  I-GPODE variants on the bouncing ball dataset without velocity observations. As shown in Table 4, using our structured  latent variable formulation (I-GPODE-L-S) significantly outperforms modeling in the data space directly (I-GPODE).
> > > (ii) In Section 5.2.5 (Table 5), we show that adding a global latent variable to the model in order to infer the charging information
> > > (I-GPODE-S with global latent variable vs. I-GPODE without global latent variable) increases performance on the particle dataset.
> > > - We thank the reviewer for the reference paper and agree that post-hoc calibration of neural networks have shown remarkable success in many applications. In this work, however, our goal is to introduce a new model class and discuss its advantages and drawbacks over similar formulations. We aimed to draw conclusions based solely on model family (e.g. GP vs NN) and studied their behavior in a variety of different scenarios (e.g. with/without velocities or charge variables). To avoid potential confounders, we deliberately employ well-established practices instead of fine-tuning each individual model. While we acknowledge that both NN- and GP-based variants of our framework can further be improved with additional measures, such as GP kernel engineering or NN post-hoc uncertainty calibration, we believe such measures lie outside the scope of this study.
> > > - We thank the reviewer for this clarification will revise the abstract accordingly.

---

### Official Review · Reviewer_F5hb · 2022-06-29

**Rating:** 9
**Confidence:** 4
**Soundness:** 4 excellent
**Presentation:** 4 excellent
**Contribution:** 4 excellent

**Summary:**

The authors propose a novel uncertainty-aware modeling of continuous-time dynamics of interacting objects. They model the trajectory of interacting systems using latent GP-ODEs by incorporating the interaction term explicitly into the differential function, and use inducing points to scale their variational inference. They show on several datasets that their method outperform other state-of-the-art approaches, that either don't model interactions between the systems explicitly or aren't uncertainty-aware.

**Questions:**

I really enjoyed the paper and the proposed model, and have only a few minor points.

In tables 4 and 5, why is I-GPODE-L performing so poorly (I-GPODE-L-S > I-GPODE >> I-GPODE-L)?
The improvement of I-GPODE-L-S over I-GPODE is discussed but I would like to have some explanation on why I-GPODE-L is performing worse than I-GPODE.

In Figure 2 and Appendix Figure 5, why do the shaded lines not follow the inferred trajectory?
The links to citation and figures are not clickable, I am not sure if this was intentional or not. On line 123, you also forget to include the section number in "Finally, please see Section for an investigation"

Equation 6 with generalization to multiple sequences in the appendix
In the Appendix table, I would recommend highlighting the best result in bold, as is done in the main paper. The tables in the appendix are quite hard to read.

I would recommend citing [Learning interpretable continuous-time models of latent stochastic dynamical systems, Duncker et al.., ICML 2019], that is the first paper proposing GP-SDE. This work was prior to [Scalable Inference in SDEs by Direct Matching of the Fokker–Planck–Kolmogorov Equation, Solin et al., NeurIPS 2021] and differed in their Inference algorithm.

**Limitations:**

I don't see any potential negative societal impact of their work.

**Strengths And Weaknesses:**

This is a great paper, as it is the first proposed method for uncertainty-aware modeling of continuous-time dynamics of interacting objects. This is an important contribution to the field of latent variable modeling. The paper is very clear, well-written, and results support the claims made in the paper.

---

> ### Author Response · Authors · 2022-08-02
> **Our response to reviewer comments**
>
>
>
> - **Question:** *"In tables 4 and 5, why is I-GPODE-L performing so poorly (I-GPODE-L-S > I-GPODE >> I-GPODE-L)?"*
> -  **Our response:** Thanks for this interesting question. In vanilla I-GPODE, all states directly
> go into the likelihood and we inject structure knowledge in I-GPODE-L-S. We empirically
> observed that missing both of these leads to training pathologies. More specifically, the
> latent states that are not matched with the data, if they do not follow any structure either,
> oscillate a lot, which in turn hampers their optimization.
>
> $ $
>
>
> - **Question:** *"In Figure 2 and Appendix Figure 5, why do the shaded lines not follow the inferred trajectory?
> The links to citation and figures are not clickable ... "*
> -  **Our response:**  Thanks for warning about the broken links and sorry for any inconvenience,
> we will take care of them in the camera ready.
> Concerning the uncertainties; we were also surprised to see that the observed trajectories do
> not always lie in the shaded regions (especially for I-GPODE model). We think this can
> attributed to the following two reasons: First, the bouncing ball set-up is challenging to
> the GP model family, since the smoothness assumption of the RBF kernel cannot perfectly
> capture the collisions; the model predicts smoother trajectories compared to neural net
> based alternatives (see Figure 2, I-GPODE, ball-3, x-vel plot). Second, we also believe the
> the lack of abundant data leads to high model bias for both I-GPODE and I-NODE (see the
> incorrectly identified collision moments in Figure 2, both panels, ball-1, y-pos plot).
>
> $ $
>
>
>
> - **Question:** *"I would recommend citing Learning interpretable continuous-time models of latent stochastic dynamical systems, Duncker et al.., ICML 2019] ... "*
> -  **Our response:** Thanks for the reference, we will include it in the final version in the paragraph
> "GPs for ODEs" in Section 4.

---

> > ### Comment · Reviewer_F5hb · 2022-08-03
> > **Re: Our response to reviewer comments**
> >
> > I thank the authors for their answer and clarifications!

---

### Official Review · Reviewer_QEA6 · 2022-07-01

**Rating:** 7
**Confidence:** 3
**Soundness:** 4 excellent
**Presentation:** 4 excellent
**Contribution:** 3 good

**Summary:**

The authors introduce a GP interaction model (I-GPODE) which is capable of modelling the continuous time dynamics of a set of interacting objects, whilst providing a robust quantification of uncertainty using latent GP-ODEs. The kinematics of each individual object and the interactions between all objects present are modelled separately; this decomposition allows the complex continuous-time dynamics to be handled by the model. The work leverages recent advances in neural ODEs and efficient sampling from GPs using pathwise updates. Results presented are better or comparable than competing methods in terms of MSE, but considerably better than all competing methods in terms of ELL.

**Questions:**

As mentioned in the Weaknesses section above, how are the performance metrics and computational performance affected by an increasing number of objects? What is the upper limit in terms of what is feasible to model? I think a relatively brief set of results considering a scenario where $N_a \gg 5$ would make a nice addition to the results section.

**Limitations:**

There is some discussion of the limitations of the work (e.g. overfitting occurring if the dimensionality of the ODE state is larger than necessary), however I think this could have been expanded a little. There is no discussion of negative societal impact, although I don’t feel that this is an issue for the subject matter in question.

**Strengths And Weaknesses:**

 ### Strengths
The I-GPODE introduced in this work is a novel contribution, addressing the problem of *uncertainty-aware* modelling of interacting object dynamics. The authors do benchmark against competing models which can also be applied to this problem setting, and some of these approaches do outperform the I-GPODE in terms of MSE in some cases, however the ELLs presented throughout the results section make a very compelling case that the I-GPODE significantly improves upon existing work in terms of robust uncertainty quantification in this setting.

The introduction, background and related work sections give a very good overview of where this work sits in terms of the wider research area; Table 1 also helped lend a lot of context. The generative model itself and the variational inference scheme employed are both laid out in a clear and concise manner; Figure 1 does a good job of quickly outlining the key ideas before you even get to the main contributions in Section 3.

The submission is well written and presented, easy to follow and technically sound. Overall, this is a great piece of work which addresses an open problem, therefore I recommend acceptance.

### Weaknesses
Whilst there is discussion of how the model complexity scales with the number of evaluation time points, I think some discussion of how the number of objects considered affects both the performance metrics and the computational feasibility of this approach (e.g. computational complexity and/or runtimes) would have been interesting to see.

---

> ### Author Response · Authors · 2022-08-02
> **Our response to reviewer comments**
>
>
> - **Comment:** *"There is some discussion of the limitations of the work ..., however I think this could have
> been expanded a little. There is no discussion of negative societal impact"*
> -  **Our response:** Please see the "limitations" paragraph above
>
>
> $ $
>
>
> - **Comment:** *"How are the performance metrics and computational performance affected by an increasing number of objects?"*
> -  **Our response:** Thanks for this constructive feedback.
> *Computational Performance:*
>         Evaluating the differential function costs $O(M^2 (V+E))$ for a graph with $V$ vertices and $E$ edges and $M$ inducing points.
>         The term $O(M^2V)$ stems from the independent kinematics, the term $O(M^2E)$ from the interaction effects.
>         As a consequence, forward integration takes $O(TM^2 (V+E))$ time, where $T$ is the number of time points.
> *Performance Metric:*
>          Based on your feedback, we performed additional bouncing ball experiments with more balls. Since the original experiment setup was designed for 3 balls, we double the box width/height (otherwise an excessive number of collisions take place, which impedes learning). The results in the below table are consistent with our main paper: *(i)* I-GPODE achieves better ELL than I-NODE while the opposite holds for MSE, *(ii)* While I-GPODE and I-BNODE achieve comparable ELLs, the latter leads to much bigger absolute errors. Kindly note that I-BNODE training on a dataset with 10 balls did not converge even though we repeated the training twice within the author response period.
>
> | Dataset| I-GPODE MSE |  I-GPODE ELL | I-NODE  MSE |  I-NODE ELL | I-BNODE MSE |  I-BNODE ELL |
> | :---: | :-----------: | :---: | :-----------: | :---: | :-----------: | :---: |
> | 3-balls |5.45 | -41 | 4.98 | -150 | 9.91 | -38 | \\
> |5-balls | 6.16 | -40 | 6.48 | -135 | 11.5 | -35 | \\
> |10-balls | 11.6 | -45 | 10.9 | -281 | - | - |

---

> > ### Comment · Reviewer_QEA6 · 2022-08-03
> > **Re: Our response to reviewer comments**
> >
> > I thank the reviewers for addressing both of my questions and particularly appreciate both the theoretical and experimental results regarding my comment on scalability with respect to the number of objects. This is a novel, interesting work which I believe meets the standard for acceptance, therefore I stand by my current rating.

---

### Official Review · Reviewer_RykX · 2022-07-12

**Rating:** 6
**Confidence:** 3
**Soundness:** 3 good
**Presentation:** 3 good
**Contribution:** 2 fair

**Summary:**

This manuscript proposes a method to infer the dynamics of interacting objects while incorporating uncertainty estimates. They use a gaussian process based ODE method, while modeling the dynamics for a set of interacting variables as a combination of independent and interacting terms. The authors also consider the case where they have partial observations of the system. They show the utility of their method on the bouncing balls and charges datasets.

**Questions:**

Please include figures such as in Figure 2 for the comparison between GPODE and I-GPODE. It is unclear why the ELL is much worse using GPODEs.

While the test in Section 5.2.3 is well performed, it would be good to list how I-GPODE does on three interacting objects as well as on the individual objects.

It would have been good to show the results of this method in a real-world scenario.

**Limitations:**

The authors have a limited discussion of the limitations and potential negative societal impact of their work.

**Strengths And Weaknesses:**

The paper is quite clearly written. The authors detail out the method and the relationship to other methods succinctly. The method they develop is an extension to the GP-ODE method, with additional constraints to take into account the interactions and independent variables. As such, it has limited originality. This also limits the significance of the paper.

While the authors show the advantages of using their method as opposed to the GP-ODE (lower MSE and better ELL) on both datasets, it is unclear why the I-GPODE would have a better ELL than the GPODE, since the main change that the authors are proposing is an additional structure. Moreover, their method does not seem to be as accurate as the GDE (I-NODE) method in the MSE (Tables 3 and 5), which we also see in Figure 2. This dampens the enthusiasm for the paper.

The test performed in Section 5.2.3 is a good one - they first learn the dynamics of three interacting objects, then simulate just one object, and see that competing methods do not learn the individual dynamics as well. This may be because the authors are explicitly learning the individual as opposed to the interactive dynamics.

---

> ### Author Response · Authors · 2022-08-02
> **Our response to reviewer comments**
>
> - **Comment:** *"... The method they develop is an extension to the GP-ODE ... As such, it has limited originality"*
> -  **Our response:** Please see the "novelties" paragraph above.
>
> $ $
>
> - **Comment:** *" ... it is unclear why the I-GPODE would have a better ELL than the GPODE ... Moreover, their method does not seem to be as accurate as the GDE (I-NODE) ... "*
> - **Our response:** Table 2 indeed reveals that decomposing the dynamics alone while keeping
> everything else the same leads to significantly increased performance than standard GP-ODE.
> We believe it implies that our structured differential function is a very useful inductive bias.
> Please see the "MSE" paragraph above concerning your second point.
>
> $ $
>
> - **Comment:** *"The test performed in Section 5.2.3 is a good one - they first learn the dynamics of three
> interacting objects, then simulate just one object, and see that competing methods do
> not learn the individual dynamics as well. This may be because the authors are explicitly
> learning the individual as opposed to the interactive dynamics."*
> -  **Our response:** Thanks for appreciating the experiment.
>         Our main goal in this experiment is to see which function approximator (GP, NN, BNN) disentangles individual effects from interactions the best.
>         In other words, the learning is performed with the same decomposed differential functions (given in eq.5), where we approximate $\mathbf{f}_s$ and $\mathbf{f}_b$ with GPs, NNs, or BNNs.
>         The results indicate that only GP approximation achieves successful disentanglement.
>         We finally acknowledge that the legend of Figure-3 might be misleading, and we will replace all the entries (e.g. from GPODE to I-GPODE).
>
> $ $
>
> - **Comment:** The authors have a limited discussion of the limitations and potential negative societal
> impact of their work.
> - **Our response:** Please see the "limitations" paragraph above.
>
> $ $
>
> - **Question:** *"Please include figures such as in Figure 2 for the comparison between GPODE and I-GPODE. It is unclear why the ELL is much worse using GPODEs."*
> -  **Our response:** We thank the reviewer for this  question.
>         Based on it, we analysed the results for GPODE in more detail and found that the worse ELL is caused by the missing inductive bias of GPODE, which prevents the model from learning, i.e. the GP stays at the prior. This also holds with reduced/increased number of inducing points as well as simpler ODE solvers such as Euler's method. For a more detailed explanation, please see the "novelties" paragraph above.
>
> $ $
>
>
> - **Question:** *"While the test in Section 5.2.3 is well performed, it would be good to list how I-GPODE
> does on three interacting objects as well as on the individual objects."*
> -  **Our response:**  Table 9 in the supplements shows the error when we train our I-GPODE on a bouncing ball dataset with three balls and test the trained model on a dataset with single ball. As can be seen, the error diminishes compared to Table 2 (the error on a test dataset with three balls). We find this rather expected as the dataset with one ball does not involve any collisions, making the dynamics much easier and predictions much more accurate.
>
> $ $
>
>
> - **Question:** *"It would have been good to show the results of this method in a real-world scenario."*
> -  **Our response:** We agree with the importance of showing results in a real-world scenario. However, we think that the initial investigation of a new methodology that operates in a complex environment with chaotically interacting objects can be made more accurately when the environment is fully controlled. Once its characteristic properties are unraveled, well-engineered implementations of the new methodology into real-world setups would give more informative results about its success. We treat the latter enterprise seriously enough to leave it to a future dedicated study. Provisionally, it could be the behavior prediction module of an autonomous driving pipeline, which includes i) perception of neighboring vehicles, ii) behavior prediction of neighboring vehicles, iii) trajectory planning of the ego vehicle, iv) optimal control of the ego vehicle towards the planned trajectory. An open-loop evaluation can be done on the well-known NG-SIM data set, for instance adopting the setup in [Y. Tang and R. Salakhutdinov, Multiple Futures Prediction, NeurIPS, 2019]. Further closed-loop tests could be done on the Carla simulator [Dosovitskiy et al., CARLA: An Open Urban Driving Simulator, CoRL, 2017]. A large application family could be model learning in Dyna-style multi-agent reinforcement learning. The corresponding real-world use cases include multi-robot manipulation, portfolio management, and social dilemma analysis in econometrics.

---

> > ### Comment · Reviewer_RykX · 2022-08-08
> > **Author Response**
> >
> > I thank the authors for their response. I would encourage the authors to include highlight the differences between GP-ODE and their approach in the paper itself, including a discussion of the sort included in the rebuttal. I am willing to increase my score given the authors' response.

---

> > > ### Author Response · Authors · 2022-08-09
> > > **.**
> > >
> > > We thank the reviewer for the concrete suggestion and increasing their score. We will highlight the differences with GPODE more rigorously in the camera-ready version.

---

### Author Response · Authors · 2022-08-02
**Initial reply to all reviewers**


We thank all reviewers for their constructive feedback and for their time in creating well thought out reviews. Here, we would like to respond to shared criticisms: the lack of novelty and significance (**reviewers RykX and gdCd**), higher MSEs compared to neural net based alternatives (**reviewers RykX and gdCd**), as well as the missing discussion on the limitations and societal impact of our work (**reviewers RykX and QEA6**). Other reviewer comments are addressed in respective boxes.
We will update the camera-ready version accordingly, incorporating all points that come up during the discussion.

---

> ### Author Response · Authors · 2022-08-02
> **Comments to all reviewers**
>
>
> ### Novelties
> Our approach differs from standard GPODE in three major respects, which we believe are important contributions to continuous-time dynamics learning literature:
> - Standard GPODEs model interactions by allowing the time differential to take the whole state vector $\textbf{H}(t)$ as input and to learn one independent GP for each latent dimension. However, using a shared latent space over all objects without our specific structure, comes at a high price: (i) number of objects is fixed, (ii) not invariant to ordering, (iii) the size of latent space grows linearly with the number of objects. The latter is challenging for GP modeling since common kernels suffer under the curse of dimensionality. For example, on the bouncing ball dataset with $N_a$ balls and $D$ latent states per object, GPODE needs to learn a latent function with $N_a D$-dimensional inputs and outputs.
> 	In contrast, I-GPODE needs only to learn two functions, the independent kinematics $\textbf{f}_s$ and the interaction function $\textbf{f}_b$, whose input sizes scale independently of $N_a$.
> 	Our experiments (Table 2) indicate that learning interaction dynamics without the strong inductive bias of our model is difficult and the GPODE model chooses to stay at the prior instead leading to deteriorated MSEs and ELLs.
> - We infer object-specific latent variables that modulate the dynamics. In turn, this allows our model to disentangle the dynamics evolution (time differential function) from static object properties (e.g. charge information).
> - We perform the learning in a latent space, where the initial value of a trajectory is given by an encoder, leading to the following two advantages: First, the dynamical system and the data points may live in different spaces, which enables learning from high-dimensional sequences. Second, our formulation can accommodate for observational noise caused by the measurement system.
>
> ### Mean squared error (MSE)
> As shown in Tables 2 and 11, I-NODE achieves lower MSE than our I-GPODE, particularly for no and low-noise regimes.
> With increased observation noise, this performance gap diminishes and eventually becomes non-significant (see the "higher noise" rows in Table 11).
>
> Nevertheless, having defined a probabilistic framework, we do not aim to outperform deterministic variants of our methods in frequentist metrics such as MSE.
> Our main claim is that I-GPODE delivers significantly better calibrated prediction uncertainty estimates than I-NODE, while maintaining on par with it in terms of prediction accuracy. This fact is demonstrated by the superior ELL scores for our I-GPODE than all baselines reported in all tables. This outcome supports our hypothesis that GPs are instrumental for uncertainty-awareness in modeling the behavior of interacting objects from partial observations.
>
> We would like to highlight that uncertainty calibration is treated in the machine learning research community as a fundamental research question on its own [V. Kuleshov et al., Accurate Uncertainties for Deep Learning Using Calibrated Regression, ICML 2018]. An accurate estimate of prediction uncertainty is a central building block of many learning problems, e.g. out-of- domain detection or active learning. It is also essential for development of downstream functionalities in safety-critical setups, such as calling doctor’s attention when a patient is diagnosed with high uncertainty or handing over control to a human driver when the environment of the ego vehicle is sensed to be chaotic.
>
>
> ### Limitations and societal impact
>
> We will add the following three paragraphs to the Discussion section of the camera-ready version.
> - *Modeling limitations:* The capacity of our GP is limited by the choice of the kernel, e.g., the RBF kernel
> 	assumes that the dynamics are stationary. While our model formulation can be combined with arbitrary kernel functions and it is possible to increase the kernel expressiveness, e.g by building composite kernels or coming up with hand-crafted features, these approaches are often time-consuming and lead to highly parameterized kernels that are difficult to learn.
> - *Approximation errors:* Our posterior inference scheme is inaccurate due to our variational framework, approximation errors that accumulate in time during future prediction, and numerical errors caused by numerical integration of an ideally continuous dynamics and its solution.
> - *Broader impact:* In our work, we propose a methodological contribution which is blind to specific data distributions. Its potential and unforeseeable side-effects in fairness-sensitive or safety-critical applications need to be investigated in a dedicated study.

---

### Meta-Review · Area_Chair_Sb9g · 2022-08-25

**Recommendation:** Accept
**Confidence:** Certain

**Metareview:**

The paper proposes a Gaussian-process approach to modelling nonlinear dynamical systems with multiple interacting objects.

The main strength of the paper is the empirical performance in terms of uncertainty quantification, backed up by clear writing and logical experimentation. Some reviewers were concerned about novelty, but I consider that additional structure proposed here to be a significant and interesting improvement on the GPODE.

One reviewer raised a question about computation complexity which was covered in the discussion. Please add this discussion to the manuscript.

I agree with reviewer gdCd that "The claim in the first line in the abstract that "for the first time uncertainty-aware modelling of continuous-time dynamics of interacting objects" is false" please modify the manuscript appropriately.

Other than those edits, this paper seems to have excited the reviewers and I'm in agreement that this presents an interesting approach that I expect others to pick up and build on.



**Award:**

No

---

### Decision · Program_Chairs · 2022-09-14

Accept